# Linear and Non-Linear Modelling Methods for a Gas Sensor Array Developed for Process Control Applications

**DOI:** 10.3390/s24113499

**Published:** 2024-05-29

**Authors:** Riadh Lakhmi, Marc Fischer, Quentin Darves-Blanc, Rouba Alrammouz, Mathilde Rieu, Jean-Paul Viricelle

**Affiliations:** Mines Saint-Etienne, Univ Lyon, CNRS, UMR 5307 LGF, Centre SPIN, F-42023 Saint-Etienne, France; marc.fischer@emse.fr (M.F.); quentin.darves-blanc@emse.fr (Q.D.-B.); rouba.alrammouz@emse.fr (R.A.); rieu@emse.fr (M.R.); viricelle@emse.fr (J.-P.V.)

**Keywords:** sensor array, Power to X, multivariate analysis, PLS, ANN

## Abstract

New process developments linked to Power to X (energy storage or energy conversion to another form of energy) require tools to perform process monitoring. The main gases involved in these types of processes are H_2_, CO, CH_4_, and CO_2_. Because of the non-selectivity of the sensors, a multi-sensor matrix has been built in this work based on commercial sensors having very different transduction principles, and, therefore, providing richer information. To treat the data provided by the sensor array and extract gas mixture composition (nature and concentration), linear (*Multi Linear Regression—Ordinary Least Square* “MLR-OLS” and *Multi Linear Regression—Partial Least Square* “MLR-PLS”) and non-linear (*Artificial Neural Network* “ANN”) models have been built. The MLR-OLS model was disqualified during the training phase since it did not show good results even in the training phase, which could not lead to effective predictions during the validation phase. Then, the performances of MLR-PLS and ANN were evaluated with validation data. Good concentration predictions were obtained in both cases for all the involved analytes. However, in the case of methane, better prediction performances were obtained with ANN, which is consistent with the fact that the MOX sensor’s response to CH_4_ is logarithmic, whereas only linear sensor responses were obtained for the other analytes. Finally, prediction tests performed on one-year aged sensor platforms revealed that PLS model predictions on aged platforms mainly suffered from concentration offsets and that ANN predictions mainly suffered from a drop of sensitivity.

## 1. Introduction

The development of green energies brings with it the problem of their intermittency. One solution is to interconnect electricity, gas and heat networks. In this way, surplus electrical energy produced at times of low household consumption could be stored in chemical form by producing H_2_ through electrolysis [1]. H_2_ thus produced can be used for mobility applications, inserted in limited quantities into the domestic gas network, or used to capture CO_2_ (produced during natural gas combustion) that will be transformed into CH_4_ via the methanation reaction [2].

Nowadays, there are various processes for producing dihydrogen and methane. These include biogas reforming, pyrolysis or pyro-gasification of biomass, methanation (chemical or biological), and electrolysis. The molecules that are mainly involved in those processes (if we do not consider the different hydrocarbonated molecules that belong to the biomass) are: CH_4_, CO, CO_2_, H_2_, and H_2_O. 

As the development in this Power to X area intensifies, issues related to process safety (CO or H_2_ leakage, possibly concomitantly) and process control (measurement of CO_2_ or CH_4_ concentrations, possibly in the presence of H_2_) are emerging. 

To monitor and control industrial processes, monitor stack emissions or detect leaks, some companies are equipped with analyzers installed as close as possible to the part of the process to be characterized (reactors, pipes, unit operations). The nature of these analyzers depends on the gases to be detected. Among the analyzer technologies used, we can find infrared analyzers, chromatography, Raman spectroscopy-based analyzers, and photoacoustic analyzers [3,4,5]. However, analyzers of those types cost several tens of thousands of euros per unit. As a result, equipping a production unit with several analyzers can be quite costly. An alternative solution would be to measure gaseous composition using a multi-sensor platform, the cost of which would be reduced by a factor of 10 to 50 compared with an analyzer. After automatic sampling (using mass flowmeters), the gaseous solution to be analyzed may have to be diluted to ensure that the concentrations to be analyzed are compatible with the sensors’ detection ranges, and reduced to atmospheric pressure, as most sensors have a restricted operating range in terms of pressure.

The well-known problem of sensors is their lack of selectivity [6,7,8,9]. To overcome this problem, several approaches exist which have given rise to specific research. One of the methods currently used consists of modifying the composition of the sensor. This can be achieved by adding a selective sensitive layer that responds to one target gas [10,11,12,13] or by integrating a filter that will block access to the sensor’s reaction sites to certain gases, similarly to the work reported by Gao et al. [14]. Another method used to achieve good selectivity consists of using sensor arrays. This technique may be used as an alternative to the first one. In this case, no modification of the sensor’s composition is made but signal treatment based on multivariate analysis enables the identification of the analytes’ nature and concentration due to the increased size of data collected by the different sensors of the array [15,16]. These arrays can use different sensors based on one transducing principle, or can group sensors with different transducing principles (arrays of MOX sensors [17,18,19], arrays of electrochemical sensors [20,21], and so on).

To achieve good prediction of a gas mixture composition, a model has to be built based on the results obtained in a first phase (“training” phase) before being validated through an independent dataset. Prediction models can be multilinear. In this case, the model will consist of a matrix. Concerning the linear methods classically used for modeling purposes, PCA calculates matrices to project variables into a new space, using a new matrix to show the degree of similarity between variables. This method is classically used in the sensor field to classify sensor signals of electronic noses into odor types [22] which can be useful in the food industry or in testing indoor/outdoor air quality, for example. However, this method is not relevant for the identification of both natures and concentrations of gases as is required in the case of process control. For this purpose, Multi Linear Regression—Ordinary Least Square (MLR-OLS) presents the advantage of being quite simple to implement [23,24]. Modeling performances are interesting for calibration and for concentration predictions when the explanatory variables (models inputs including the sensor signals) are not correlated with each other. In this case, the PLS model is more appropriate. Indeed, this method is very effective, especially when the sensor signals are linear in their detecting range [25]. For example, in a study performed by Karami, Rasekh [26], an e-nose with MOS sensor was used to detect oil oxidation. The reliability of the PLS method in detecting this phenomenon was the most interesting among the tested methods, and was assessed at 100%. 

On the other hand, it seems interesting to consider non-linear models which can be more effective in the case of strong non-linearity in the input sensor signals. More or less complex artificial neural network (ANN)-based models will, for example, allow gas mixture composition prediction [27] and also gave good results when used with e-noses to assess the quality of products. As we will explain in the following sections, preliminary tests consisting of exposing sensors to mono-analyte gas compositions gave, in almost every case, linear responses in the targeted detection range. Non-linear models like neural networks are suitable for linear behaviors but, due to the fact that the extraction of the model parameters is based on reaching local minimums [28], the extracted parameters will not always correspond to the most adequate model. 

Predictions can be biased due to numerous problems: different types of transitory phenomena (temperature or pressure evolution), ageing of the electronics for signal treatment, and drift/ageing of the sensors constituting the platform. Different reversible or irreversible processes may cause short- or long-term sensor drift. Reversible damage, which results in short-term drifts, can result from condensation of chemical vapor on the active surface of the sensor, physical adsorption of chemical compounds, or evolution of ambient atmospheric conditions (temperature, humidity, influence) [29]. Irreversible damage can result from a brutal phenomenon, for example, the poisoning of MOX or electrochemical sensors with sulfur compounds or from a continuous evolution over time. This last case can be due to the evolution of the electronic components’ dedicated signal treatment or from an evolution of the sensor’s active materials (electrodes, semiconductor oxides, and heating element) due to surface chemical reactions or degradations due to mechanical stresses [30]. 

In this paper, two linear models and several neural network-based models with and without a hidden layer will be compared in terms of predictive capabilities. Single analytes and binary mixtures will be considered for developing and testing the models. In addition, the article also includes the study of the pertinence of the concentration predictions after one year of continuous use of the platform under controlled environmental conditions. In this case, predictions will be affected by irreversible damage due to continuous ageing of electronics or sensors. Comparison of the prediction performances of linear (PLS) and non-linear (ANN) methods, initially and after one year of continuous platform use, will be carried out. Indeed, one of the major objectives of this work will be to evaluate the evolution over time of the performance of commonly used linear and non-linear models for predicting gas concentrations in the simple case of binary mixtures.

## 2. Materials and Methods

### 2.1. Sensor Choice

The choice of sensors for the multi-sensor platform was made according to well-defined specifications. The first one concerned the targeted gases: CO_2_, H_2_, CH_4_, and CO, which are the most common gases in Power to X-linked processes. Additional temperature and humidity sensors are also needed since those two parameters will vary during the detection phase and since they constitute potential influences. The desirable gas concentration detection ranges are both a function of the gas concentrations that can be measured at specific points of the processes and the possibilities offered by commercial sensors. In order to detect traces of CO in H_2_ (methanization) or leakage of CO, the targeted detection range expected for CO runs from a few ppm to some hundreds of ppm. For H_2_, requirements relate to its monitoring in a process and the detection of H_2_ leaks. For safety reasons and for reasons of sensor range limitations, the targeted detection range for H_2_ has been limited from a few hundred ppm to 1%. For CO_2_ and CH_4_ gases, the requirements concern only the gas concentration monitoring at different steps of the process. Ideally, sensor detection ranges should be from a few hundred ppm up to 100% but, as will be shown, the sensors’ upper limit for those gases does not exceed a few tens of %.

In order to address the specifications, 5 commercial sensors were selected for this project. These ones were deliberately chosen with very different operating principles in order to maximize the versatility of the associated responses. 

Prior to selecting the sensors, a study was carried out to determine the sensor technologies that could be used to detect the gases of interest to the project (CO, CO_2_, CH_4_, and H_2_). The results are summarized in Table 1.

Sensor technologies can be classified into two families: chemical and physical sensors. The first family is based on the change of an electrical output characteristic’s value due to a chemical reaction. CO_2_ is a weakly reactive molecule. Therefore, chemical sensors (MOX, catalytic, electrochemical) are not the most effective to detect it. Commercially, two physical sensor technologies exist for the detection of CO_2_: Non-Dispersive Infrared (NDIR) sensors and photo-acoustic sensors. The second technology is relatively recent and only a few constructors propose it. The photo-acoustic effect is based on the absorption (by target molecules) of a modulated (or pulsed) light beam. As the molecules de-energize through collisions, they generate sound waves that are detected by a condenser microphone. Yet, the technology that is mainly used for the detection of CO_2_ remains the classical infrared absorption-based one. As for the larger IR analyzers, those sensors will be based on the detection of a change of luminous intensity due to absorption of NDIR rays by the gaseous analyte. In an MOX sensor, the variation in electrical conductivity of an oxide semiconductor layer is measured as a function of the presence of chemisorbed gaseous analytes (redox interface reactions). Concerning electrochemical sensors, redox reactions are also involved but at electrode/electrolyte/gas interfaces. Those reactions will affect the interface resistance of the working electrode (mainly) and change the electromotive force measured between the electrodes. For catalytic sensors, a specific combustion (redox) reaction will occur on two alumina beads: a reference bead and another bead (for measurement) covered with a catalyst which decreases the combustion reaction temperature. Those alumina beads are traversed by a platinum wire, which is an RTD material, i.e., a material which changes resistance with temperature. Resistance variation due to analyte combustion is measured thanks to a Wheatstone bridge involving both the reference and the measurement bead resistances. In those three technologies of sensors, electrochemical reactions are involved. Due to the variety of semiconductor oxides used in MOX sensors, commercial references were found for CO, H_2_ and CH_4_ compounds. For the electrochemical sensors, a lot of references were found for CO detection and references were found for H_2_ detection also. Concerning the catalytic sensors, references found mainly concerned the detection of hydrocarbons (including CH_4_) and H_2_.

From the references identified, the next step was the selection of the sensors that will be used in this work. This selection was performed according to many criteria: respect of the process linked specification (presented earlier), the versatility of the sensor signal expected (it was important to avoid collinearity between the sensor responses), and low number of interferents (especially humidity and temperature interferents). 

Table 2 lists the name, model, type, detection range, and target gases for each selected sensor. A digital NDIR sensor was chosen for CO_2_ measurement. This one incorporates a temperature sensor as well as a humidity sensor, so that the CO_2_ concentration signal delivered by the sensor incorporates compensation for temperature and humidity variations. Hence, our system will eventually have 7 sensors. The temperature and the humidity sensor will also be used in the sensor network as input parameters in the multi-linear models, but not in the ANN as we feared this could lead to overfitting. Unlike other sensors, the NDIR sensor is selective. Determining CO_2_ concentration will therefore not require multivariate analysis as will be the case for the other gases. The platform also incorporates two electrochemical sensors (EC-H_2_ and EC-CO) which are highly sensitive to H_2_ and CO, but not selective. Finally, a catalytic sensor (CATA) and a Metal Oxide sensor (MOX) were chosen for their sensitivity to CO, H_2_, and CH_4_.

### 2.2. Experimental Setup

A gas bench equipped with a series of flowmeters was used to generate gas mixtures of specified compositions with a total gas flow of 30 L/h (Figure 1a). In order to expose the sensor platform to single analyte gases or binary mixtures, 2 sensor platforms of 7 sensors (if we include the temperature and humidity sensors) were introduced in sealed cells such as the one shown in Figure 1b and exposed to the gas mixtures. The sensor signals were conditioned using commercial or laboratory-developed analogic and digital electronics as can be seen in Figure 2. EC-CO and EC-H_2_ sensors require a special conditioning step performed by analogic circuit boards. However, for the rest of the sensors, the conditioning is performed by the central “laboratory-made” circuit board. Finally, signals are digitized and computerized using an Arduino board for NDIR CO_2_ sensors, and a National Instruments (NI) board for the others. Once the sensor data have been collected, behavioral modeling of the platform is carried out using Excel software for the MLR-OLS method (Multi Linear Regression—Ordinary Least Square), the Python algorithm using the “PartialLeastSquares” library for the MLR-PLS method (Multi Linear Regression—Partial Least Square) and Keras/Tensorflow in Python for the neural network method. These models will be used for the final tests to predict a gas composition from the response of the sensor array immediately in the weeks after the model building and also after an ageing period of one year.

### 2.3. Test Procedure

#### 2.3.1. Role of Mono-Analyte Tests

Before exposing the sensor array to complex binary mixtures, it was important to study the sensor responses to each analyte. The goal of these “mono-analyte tests” are of different natures:-to verify the reproducibility of the sensor responses,-to check that the sensor drift is limited and close to zero,-to analyze the transfer function linking the gas concentration of the analyte and the sensor responses (linear or not),-to check that the sensor responses to the introduced analytes are sufficiently uncorrelated to have enough variability in the information collected. If two sensors respond the same way to all the analytes, they finally bring “collinear” information, which would be prejudicial for the models. Indeed, it can lead to overfitting so that the model will almost perfectly learn to match the training data but will be unable to capture the validation data.

#### 2.3.2. Sensor Network Exposure to Both Mono-Analyte and Binary Mixtures

A LabVIEW program is used both to control the flowmeters and therefore the gas mixture composition in the gas line according to time, and also to collect the data from the different sensors and gather them in a specific file. The program can be fed with a file containing sequences of gas compositions at different times that will be applied to the different flowmeters to obtain the expected gas concentration evolution as a function of time (Figure 3). Each sequence lasts between 30 and 60 min and a set of sequences is structured in the following way:

-first sequence under “base gas”: 12% O_2_/1% absolute humidity/N_2_,-several sequences including introduction of analytes alone or in binary analyte mixtures,-last sequence under “base gas”: 12% O_2_/1% absolute humidity/N_2_ to verify the return of the sensor signals to the base line, i.e., verify that the signal corresponding to the first sequence is the same as the signal at this last sequence (no drift of the sensor signals).

#### 2.3.3. Modelling Step: Behavior Model Construction

During this step, the signals collected during the phase of sensor network exposure to the different single and binary analyte mixtures will be used. They will constitute the input data from our models and will be gathered in a matrix X constituted of the elements X_ij_, where “i” corresponds to the sampled point number (which can be linked to time knowing the sampling frequency) and “j” corresponds to the sensor number: 1 to 14 (2 cells with 7 sensors each). The output matrix of the model is a table Y constituted of the elements Y_ik_, containing the concentration evolution of the four targeted analytes according to time. Here, “i” also corresponds to the sampled point number and “k” corresponds to the analyte number (from 1 to 4).

Concerning linear models, MLR-OLS [31,32] and MLR-PLS [33,34] methods were chosen. The MLR-OLS model is a modelling method in which the empirical estimation of a calibration matrix, C, allows us to use experimental sensor data (matrix X) to get a prediction matrix $\hat{Y}$ (which corresponds, here, to modelled gas concentration values). C is composed of the elements $C_{kj}$ and is determined by least squares minimization (parameter RMSE, Root Mean Square Error) between modelled values $\hat{Y_{ik}}$ and experimental values $Y_{ik}$: $RMSE=\sqrt{\sum_{i=1}^{N}\frac{{\left(y_{ik}-\hat{y_{ik}}\right)}^{2}}{N}}$. The RMSE is calculated for each analyte “k”. In the linear model, the relationship existing between the sensors’ matrix signals and the analyte concentrations is the following:(1)$$X=Y\cdot C+ones\cdot R_{0}$$
where ones is a one-column vector composed of $i_{max}$ elements (number of sampled points) whose components are all equal to 1. $R_{0}$ is a one line vector composed of $j_{max}$ elements (14 sensors here). It corresponds to the sensors’ response vector when no analyte is introduced. $R_{0}$ and the calibration matrix C determined during the training phase will constitute the parameters of the model ($C^{\prime }$ being the transpose of C).

To predict the value of the concentration matrix in the model validation phase, linear algebra is used to extract the concentration matrix $\hat{Y}$:(2)$$\hat{Y}=\left(X-ones\cdot R_{0}\right)\cdot C^{\prime }\cdot {\left(C\cdot C^{\prime }\right)}^{-1}$$

In the case where the number of predicted variables (matrix $\hat{Y}$) is rather high and at the same time the amount of information from experimental data (matrix X) insufficient, the OLS method becomes unstable because the system is undetermined. Similarly, when the number of experimental variables (sensors as predictors) is large and the amount of data used in the model-building phase is insufficient, OLS models then suffer from multi-collinearity and overfitting problems. 

The MLR-PLS model will then seek to define a model that will maximize the covariance between X and Y using latent variables. These variables replace explanatory variables (sensor signals in our case) with a more or less strong collinearity. Indeed, they constitute linear combinations of those in which the factor affected by each explanatory variable is chosen so as to maximize the covariance between the newly created latent vector and the concentration matrix. Then, the multilinear regression is not performed on the explanatory variables but on the latent variables. In this work, a version of the algorithm developed by Abdi et al. [35] in 2010 was used through Python code.

We also decided to model the relationship between the analyte concentrations and the sensor signals through a series of artificial neural networks (ANN) [36], as illustrated by Figure 4 for the case of H_2_ concentration prediction.

$f_{act,1}=tanh$ and $f_{act,2}=linear$ are the two activation functions we have chosen. When $n_{hidden}=0$, there is no hidden layer and the concentration of the considered analyte (H_2_ in the following equation) can be determined by:(3)$$Y_{H_{2},i,ANN}=f_{act,1}\left(\sum_{j=1}^{n_{sensors}}\omega_{j}X_{ij}+\omega_{0}\right)$$
where $X_{ij}$ is the value returned by the j-th sensor of the regressor combination at time i and the $\omega_{j}$ are weights to be optimised based on the training data set. $\omega_{0}$ is the weight of the bias neuron. In the presence of a hidden layer, the pollutant concentration (for instance H_2_) can be calculated using the following equation:(4)$$Y_{H_{2},i,ANN}=f_{act,2}\left(\sum_{j=1}^{n_{hidden}}\omega_{2,j}\;*\;f_{act,1}\left(\sum_{k=1}^{n_{sensors}}\omega_{1,k,j}X_{ik}+\omega_{1,0,j}\right)+\omega_{2,0}\right)$$
where the weights $\omega_{1,k,j}$ and $\omega_{2,j}$ are to be optimised based on the training data set. $\omega_{2,0}$ and $\omega_{1,0,j}$ are the weights of the bias neurons for the output and the hidden layer, respectively.

The weights are optimized through the minimization of RMSE given by:(5)$$RMSE=\sqrt{\sum_{k=1}^{n_{time}}{\left(Y_{species,ANN,k}-Y_{species,k}\right)}^{2}}$$
where n_time_ is the number of time points in the training data set, $Y_{species,k}$, is the considered experimental species concentration at the k-th time point of the training data set, and $Y_{species,ANN,k}$ is the prediction of this value by the ANN. The Gradient Descent Method with momentum [37] was employed for the optimization. After a random initialization, the weights of the neural network are iteratively adapted according to these coupled equations:(6)$$\omega_{t+1}=\omega_{t}-\varepsilon \cdot v_{t+1}$$

With:(7)$$v_{t+1}=\rho \cdot v_{t}+\left(1-\rho \right)\nabla_{\omega }RMSE\left(\omega \right)$$

ε is the learning rate which was set to 0.2, whereas ρ is the momentum constant which was set to 0.9, and $v_{0}$ is set to 0 in Keras/Tensorflow. 

If ${∆}_{t}RMSE={RMSE}_{t}-{RMSE}_{t-1}<10^{-4}$ for a duration equal to patience = 500 iterations, the optimisation is stopped as we consider that no further significant improvement can be made. The optimisation is always stopped after a maximum of 10,000 iterations. It is worth emphasizing that this is the RMSE of the training data set as the validation data set is only used after the end of the optimization to test the ability of the model to predict independent experiments. Both the regressors (sensors) and the target concentrations were normalized before the beginning of the training of the ANN so that they only take on values between 0 and 1. This is preferable if we want to use activation functions such as “tanh”.

## 3. Results & Discussions

### 3.1. Mono-Analyte Tests

Mono-analyte test results presented in Figure 5, Figure 6, Figure 7 and Figure 8 allowed us to answer all the questions raised in part 2.3 and to validate the potentialities of the sensor array to be used for predictions. The first point to validate was reproducibility. All the results presented in Figure 5, Figure 6, Figure 7 and Figure 8 were reproduced two times and, in each case, less than 10% difference was observed between the sensor responses, which is, in the case of analogue sensors, quite reasonable. Then, it was important to check that no major drifts occurred in the sensor responses, i.e., that the base line before and after the introduction of the analytes is at the same level and that the slope of the response curve according to time finally tends towards a horizontal asymptote after the gas mixture composition has been changed. This also could be validated in the results shown in Figure 5, Figure 6, Figure 7 and Figure 8. Another very important point was to check that the difference in the sensor array’s responses from one gas to another was sufficient to discriminate the target gases. It can be seen from Figure 5 and Figure 6 that sensor responses to H_2_ and CO are relatively close even though the amplitudes of the responses of the different sensors are not comparable. One element that could permit to discriminate H_2_ and CO when they are in binary mixtures with another gas is the fact that H_2_ induces a significant response in the MOX sensor, whereas for CO this is not the case. Concerning CH_4_, only the MOX sensor responds to this analyte (Figure 7), in contrast to H_2_ or CO. Mixtures of H_2_ and CH_4_ may be less convenient to identify for low concentrations of CH_4_ since the correlation between H_2_ concentration and MOX sensor is also very strong. For the three gases mentioned in this paragraph, multivariate analysis is required since two conditions are not fulfilled for those gases:

-induce a response in only one sensor;-this last sensor should only respond to this analyte.

However, this is the case for CO_2_. Indeed, CO_2_ induces a response only in the NDIR sensor (Figure 8) and, at the same time, this last sensor does not respond to the other analytes (Figure 5, Figure 6 and Figure 7—curve f). Thus, CO_2_ will not require any multivariate analysis to be discriminated. The information from the NDIR sensor will be sufficient. Therefore, in the multivariate analysis performed in the following sections, CO_2_ will not be involved.

### 3.2. Sensor Transfer Function

The sensor transfer function towards each analyte is a very important piece of information to collect to get indications of the type of models that may be the most suitable to perform concentration predictions. Indeed, if most of the sensor responses to the analytes were not linear, attempts to build a linear model to perform prediction would be unsuitable. From Figure 9, it can be noticed that most of the transfer functions are linear for all the considered sensors and analytes. Two exceptions to this statement exist. The first one concerns the response of the catalytic sensor to H_2_. In this case, the response is linear until a concentration of 600 ppm, after which there seems to be an asymptotic behavior (saturation of the sensor). The second exception concerns the MOX sensor’s response to CH_4_, for which the sensor’s response is purely logarithmic.

Despite these two non-linear transfer functions, multi-linear models remain good candidates to perform prediction of analyte concentrations because of the measurement uncertainty (mainly composed of reproducibility errors) which could be less amplified by linear models.

### 3.3. Building up Models from Training Data

To build up the different models (linear and non-linear), four specific test sequences were carried out twice each. To avoid overloading the article, these specific test sequences will be presented in Appendix A in graphical form. The sensor data from these tests were collected, formatted and used to build the models. On the raw sensor data, the transient parts correspond to both the response time of the multi-sensor array and the air renewal dynamics in the large 0.4 L cell volume (which is a function of the total flow rate of 30 L/h on the two cells). Those transient parts were removed in the data used to build up the model, leaving only the stationary parts. The data are composed of 14 columns for the 14 sensors (two cells of seven sensors) that will constitute the explanatory variables of the different models plus four columns containing the gas concentrations that will constitute the response variables. Those concentrations are accurately known since the LabVIEW program controls them through the flowmeters. From the model parameters estimated based on the training data, “modeled” analyte concentrations could be calculated using sensor responses and compared to the “real” experimental concentrations used during the training tests. The accuracy of these concentration predictions is evaluated based on the RMSE (Table 3). The only linear model enabling good concentration predictions on the training data is MLR-PLS. In Figure 10, analyte concentration predictions performed on the training dataset with this last model are represented as a function of time and compared to experimentally imposed values of H_2_, CO, and CH_4_ concentrations. It can be noticed that good prediction results are globally obtained for those three analytes. Yet, concerning the base line (concentration of 0 ppm of analytes), some false positive or negative values are obtained for CO and CH_4_. Those false positive or negative concentration prediction values are, then, expected to occur also on predictions performed on validation tests.

A series of artificial neural networks based on different sensor combinations as regressors and numbers of neurons in the hidden layer were trained and validated, as explained in *Appendix B: Results of the Neural Networks*. While the quality of the prediction of the validation data set for CO generally increases with the complexity of the ANN structures, this is not the case for CH_4_ where we can only see a decrease in the training RMSE. When it comes to CH_4_, the training RMSE sometimes increases as the complexity of the ANN is increased, which can only stem from the optimization method becoming stuck inside local minima, since the most complex structures include the less complex ones as a special case.

The results of the different models for the training phase can be seen in Table 3. The obtained RMSE are quite comparable with those obtained with the PLS method even if the prediction of H_2_ concentration seems a little less accurate for the best ANN model compared to the PLS model. These results are confirmed in Figure 11 in which we can see that the prediction results are quite comparable to those obtained with the PLS method (Figure 10).

### 3.4. Validation Tests and Comparison of Models

In this section on validation tests, the best linear model and best non-linear model are selected and used to perform prediction on data that have not been used to construct the model. The sensor data used for prediction were collected during gaseous exposure of the platform to the sequence shown in Figure 12 for the gases H_2_, CH_4_, and CO, alone or in binary mixtures. Gas concentration prediction results for PLS and ANN methods are shown in Figure 13 and Figure 14, respectively. The predictions made for the three gases are relatively satisfactory in terms of determining the nature of the gases present. However, there are still some imperfections: over/underestimation of concentrations, and false positives/negatives. Prediction of H_2_ and CO gas concentrations seems quite comparable between PLS and ANN methods even if H_2_ prediction seems slightly better with the PLS method, as is confirmed in Table 4. Concerning CH_4_, prediction results are better with the ANN method as can be seen in Figure 13 and Figure 14 and confirmed by an RMSE result of 497 compared to 755 for the PLS method (Table 4). Less effective prediction of CH_4_ with the PLS method is consistent with the fact that CH_4_ only induces a significant response on the MOX sensor, which is logarithmic. Thereby, the limits of the linear PLS model compared to ANN are shown here, even if CH_4_ concentration prediction results are consistent.

### 3.5. Data Post-Treatment

For the data predicted by the best linear and non-linear models, there are still problems that seem avoidable, particularly when the predicted concentration values are negative or positive while the actual concentrations seen by the sensor network are zero. To overcome these false negatives and false positives, a post-processing algorithm has been developed. This is based on the knowledge of sensor signal values when analyte concentrations are zero, and will be limited to concentration predictions in the latter case only, in order to avoid introducing bias. It consists of setting the analyte concentration to 0 when the signals of the different sensors constituting the platform remain under thresholds specific to each sensor. Post-treatment results are presented in Figure 15 and Figure 16 and Table 5 for MLR-PLS and ANN models, respectively. The corrections are very effective for the H_2_ analyte. In this case, both false negative and positive could be removed. In the case of the CH_4_ analyte, results are very good too, even if the algorithm seems less effective at removing the few false positives obtained. Finally, post-treatment results obtained in the case of the CO analyte are less good, especially when CH_4_ is in binary mixture with H_2_. In the latter case, in spite of the absence of CO, the proximity of the sensor responses to H_2_ and CO makes it very difficult to distinguish H_2_ from CO and CO predicted concentration is, in this case, not equal to 0.

### 3.6. Ageing of Sensors

Next, the multi-sensor platforms were maintained for a year under electrical power, atmospheric pressure, and air. At the end of the year, the gas sequence to which the sensors had been exposed for the prediction testing a year earlier was used again, to check the relevance of the prediction model. Figure 17 shows a comparison between sensor responses before ageing and after a one-year ageing period when subjected to the gas sequence shown in Figure 17a. For the EC-H_2_ (Figure 17b) and EC-CO (Figure 17c) sensors, an overall decrease in sensitivity is observed. This is more pronounced for the EC-H_2_ sensor than for the EC-CO sensor. The same trend is observed for the catalytic sensor (Figure 17d), except at higher H_2_ concentrations, where the response of the aged sensor is greater. Finally, for the MOX sensor (Figure 17e), a greater sensitivity is observed for the sensor after one year of ageing as long as H_2_ is present is the gas mixture.

Based on the sensor platform’s responses and applying the MLR-PLS and ANN models previously developed (without post-treatment), predictions of the concentration of H_2_, CO, and CH_4_ have been performed. Concerning the predictions made using the ANN model (Figure 18), different conclusions can be drawn according to the gas considered. Indeed, for H_2_, a global underestimation of the concentration is observed. When H_2_ concentration seen by the sensors is too low, predicted H_2_ concentration becomes 0. Concerning CH_4_, it seems that when this analyte is present alone in the base gas mixture, predicted CH_4_ concentration is close to 0. However, as long as H_2_ is present in sufficiently high concentration (even without the presence of CH_4_), predicted CH_4_ concentration raises. This indicates that, when trying to predict CH_4_ concentration, the ANN model will actually be more representative of the H_2_ concentration in the mixture. Finally, the best prediction results of ANN on aged sensors is obtained for CO. The concentration evolution is globally well reproduced. However, false positives in CO concentration predictions can be observed when H_2_ is present in a mixture that does not contain CO.

Then, concerning the predictions made using the MLR-PLS model (Figure 19), for the different gases, a prediction offset is observed in each case. For H_2_ and CO, a positive offset is noticed while a negative prediction offset is observed for CH_4_. Even if the prediction response amplitude seems in correlation with experimental gas concentration, the presence of this offset completely ruins prediction quality for the different gases. Application of the post-treatment algorithm described earlier is not efficient in this case because of the important offsets observed. However, those offsets could be easily removed by periodic zero calibration of the platform. In Figure 20, prediction results including offset compensation are shown. Offset-compensated H_2_ predictions reveal global underestimation of H_2_ concentration. In the case of CH_4_, concentration overestimation is reported. When H_2_ is present without CH_4_, false positives are also present. For the PLS model, best (offset-compensated) prediction results are obtained for CO, for which concentrations are quite accurately predicted. False positive CO concentration prediction is also obtained when a high concentration of H_2_ is present (without CO) in the gas mixture.

## 4. Conclusions

In a context of process developments linked to Power to X, the first goal of this work was to select commercial sensors to build a multi-sensor platform able to detect H_2_, CO, CH_4_, and CO_2_ concentrations when gases are alone or in binary mixtures while respecting specifications linked to the applications. With the sensors selected, the processing chain enabling signal treatment and digitalization was developed. Finally, the main task was to build linear (MLR-OLS and MLR-PLS) and non-linear (ANN) models capable of detecting gases in binary mixtures. The first step was training. This allowed us to train the models by estimating their coefficients and then evaluate them by their ability to reproduce the data they were trained with. This step allowed us to disqualify the MLR-OLS model, which was clearly not able to predict analyte concentrations based on the sensor signals. Another set of experimental sensor data was then used to compare the prediction performances of MLR-PLS and ANN in the case of “fresh sensor data”, i.e., data not used to build up the models. The main result was that the gas concentration predictions were quite comparable for H_2_ and CO (slightly better for H_2_ concentration predictions with the PLS method) for linear and non-linear methods and better for ANN in the case of CH_4_ concentration predictions, which is in accordance with the fact that the MOX sensor’s response to CH_4_ is logarithmic. To improve the sensor prediction performances, a post-treatment algorithm was developed to correct for the case where predictions should give 0 ppm analyte concentration. This helped to improve the RMSE, especially in the case of CH_4_.

Finally, prediction tests were performed on a sensing platform that had been aged for one year. Due to the evolution of the sensors’ responses, the quality of the predictions performed by the ANN and PLS models greatly deteriorated. While ANN predictions suffer from high underestimation of H_2_ and CH_4_ concentrations (predictions for CO concentrations being correct), the PLS model suffers from big prediction offsets. After compensation of the offset by calibration, the quality of the prediction by PLS becomes much better than ANN, even if global underestimation of H_2_ concentrations and the presence of false positives in the prediction curves of CO and CH_4_ reduce prediction quality compared to the unaged sensing platform case.

In future works on the subject, our first goal will be to test sensor signals at different stages of ageing and propose an ageing model, which would act as a second layer added to the already developed linear and non-linear models to compensate the effects of sensor ageing.

Our second goal will be the determination of more complex gas mixture compositions, such as ternary and quaternary mixture compositions, based on sensor signals. This will require the development of new models and possibly the addition of new sensors to the sensor array used in this work to enrich the information provided by the latter.

## Figures and Tables

**Figure 1 sensors-24-03499-f001:**
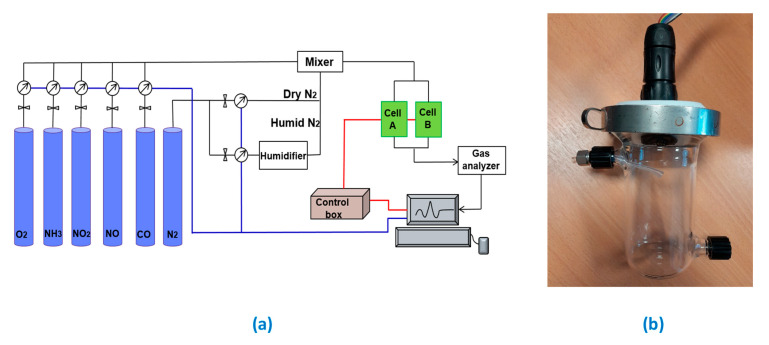
(**a**) schematic of the gas bench. (**b**) Cell containing the multi-sensor platform.

**Figure 2 sensors-24-03499-f002:**
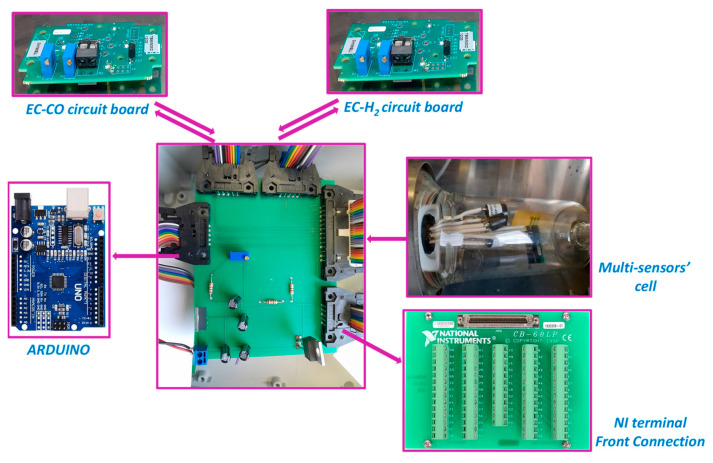
Sensors and the signal treatment chain.

**Figure 3 sensors-24-03499-f003:**
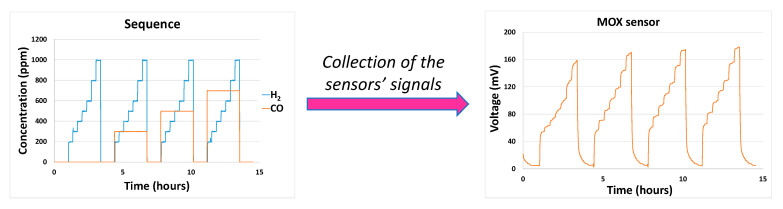
Example of gas composition sequence and gas sensor response.

**Figure 4 sensors-24-03499-f004:**
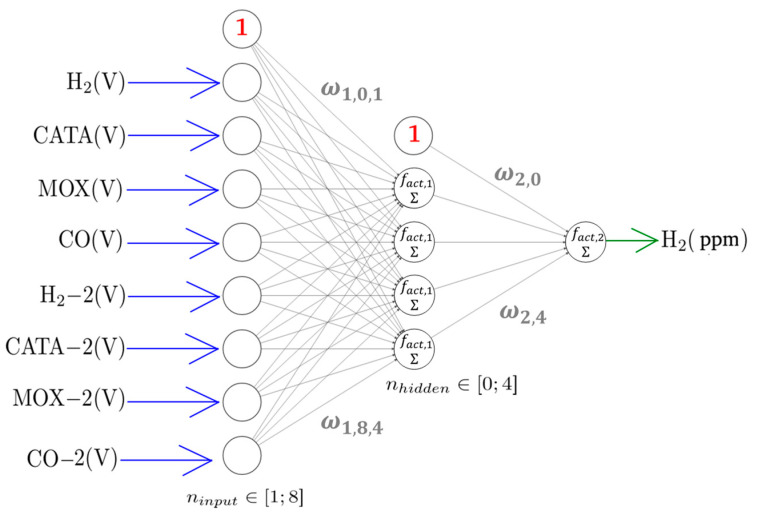
Example architecture of the artificial neural network (ANN) used for H_2_ concentration prediction.

**Figure 5 sensors-24-03499-f005:**
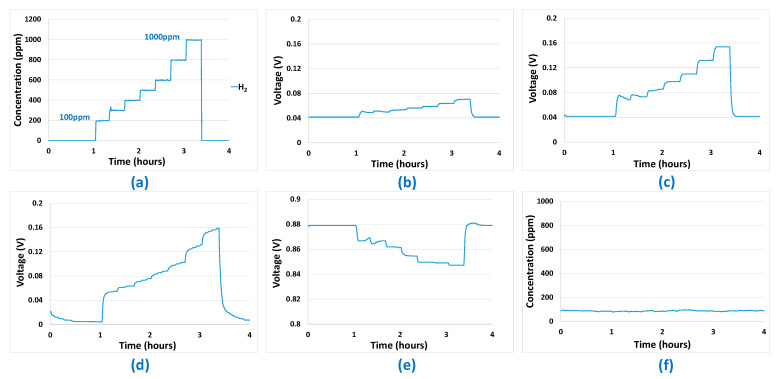
(**a**) Gas sequence with H_2_ concentrations between 100 ppm and 1000 ppm. (**b**) EC-H_2_ sensor signal. (**c**) EC-CO sensor signal. (**d**) MOX sensor signal. (**e**) Catalytic sensor signal. (**f**) CO_2_ sensor signal.

**Figure 6 sensors-24-03499-f006:**
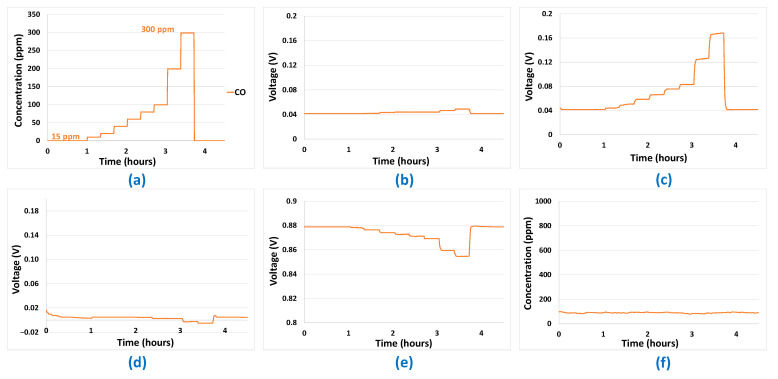
(**a**) Gas sequence with CO concentrations between 15 ppm and 300 ppm. (**b**) EC-H_2_ sensor signal. (**c**) EC-CO sensor signal. (**d**) MOX sensor signal. (**e**) Catalytic sensor signal. (**f**) CO_2_ sensor signal.

**Figure 7 sensors-24-03499-f007:**
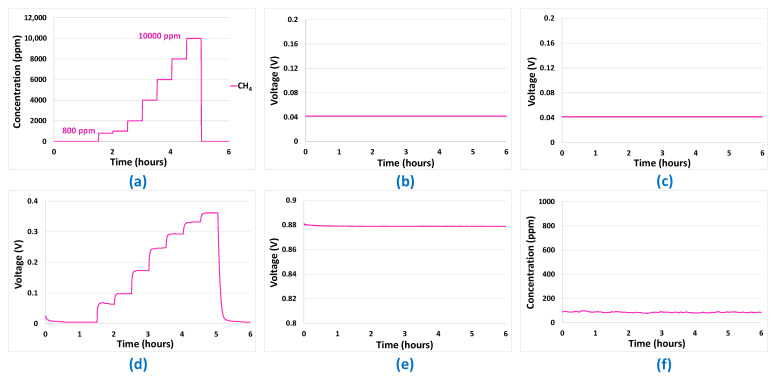
(**a**) Gas sequence with CH_4_ concentrations between 800 ppm and 10 000 ppm. (**b**) EC-H_2_ sensor signal. (**c**) EC-CO sensor signal. (**d**) MOX sensor signal. (**e**) Catalytic sensor signal. (**f**) CO_2_ sensor signal.

**Figure 8 sensors-24-03499-f008:**
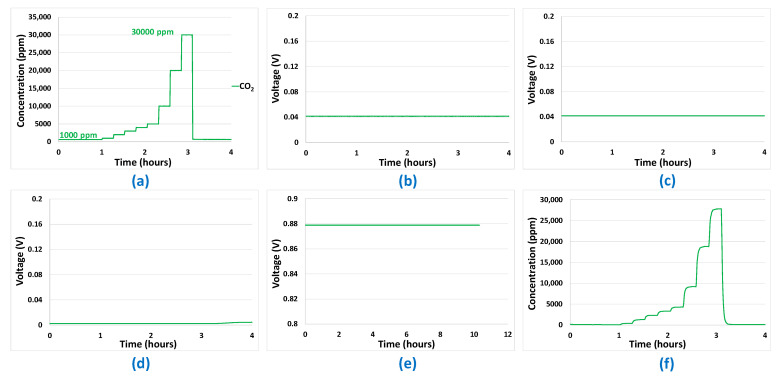
(**a**) Gas sequence with CO_2_ concentrations between 1000 ppm and 30,000 ppm. (**b**) EC-H_2_ sensor signal. (**c**) EC-CO sensor signal. (**d**) MOX sensor signal. (**e**) Catalytic sensor signal. (**f**) CO_2_ sensor signal.

**Figure 9 sensors-24-03499-f009:**
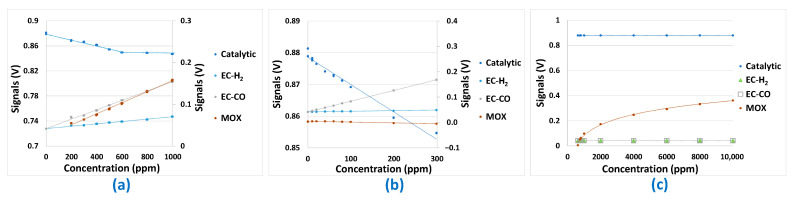
Transfer function of the sensors submitted to different concentrations of: (**a**) H_2_, (**b**) CO, (**c**) CH_4_.

**Figure 10 sensors-24-03499-f010:**
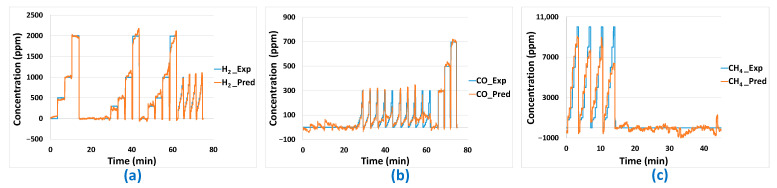
(**a**) H_2_, (**b**) CO, (**c**) CH_4_ concentration predictions based on PLS modelling with training data.

**Figure 11 sensors-24-03499-f011:**
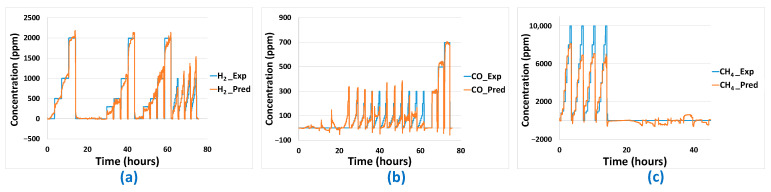
(**a**) H_2_, (**b**) CO, (**c**) CH_4_ concentration predictions based on ANN modelling with training data.

**Figure 12 sensors-24-03499-f012:**
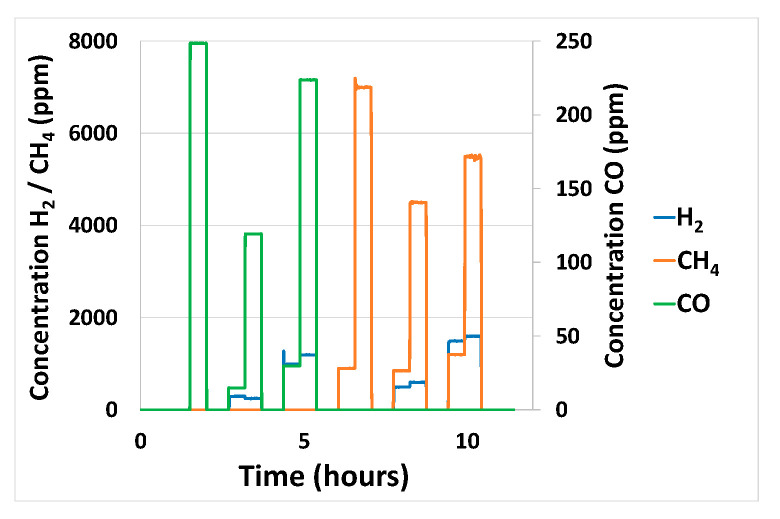
Sequence used to predict H_2_/CH_4_/CO analyte concentrations alone or binary mixtures.

**Figure 13 sensors-24-03499-f013:**
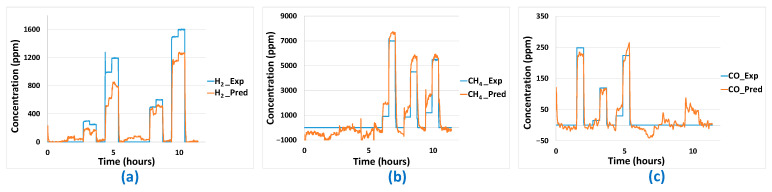
Prediction of concentrations of: (**a**) H_2_, (**b**) CH_4_, (**c**) CO by the MLR-PLS method.

**Figure 14 sensors-24-03499-f014:**
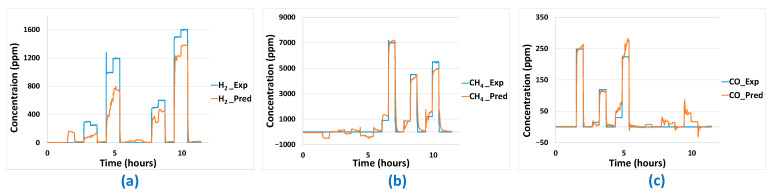
Prediction of concentrations of: (**a**) H_2_, (**b**) CH_4_, (**c**) CO by the ANN method.

**Figure 15 sensors-24-03499-f015:**
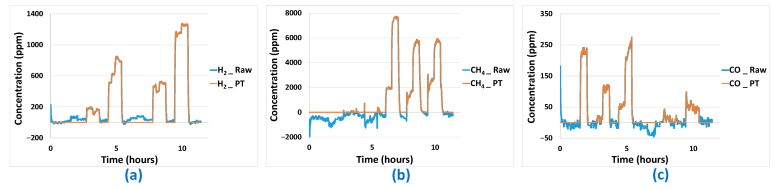
Predictions results before (_Raw) and after (_PT) application of post-treatment algorithm for PLS prediction curves: (**a**) H_2_ predictions results, (b) CH_4_ predictions results, (**c**) CO predictions results.

**Figure 16 sensors-24-03499-f016:**
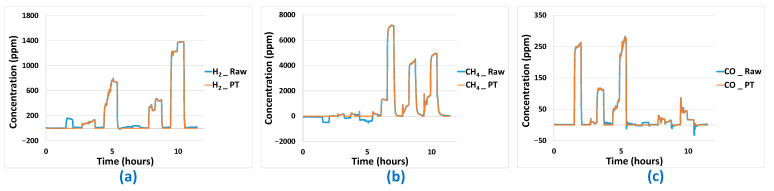
Predictions results before (_Raw) and after (_PT) application of post-treatment algorithm for ANN prediction curves: (**a**) H_2_ predictions results, (**b**) CH_4_ predictions results, (**c**) CO predictions results.

**Figure 17 sensors-24-03499-f017:**
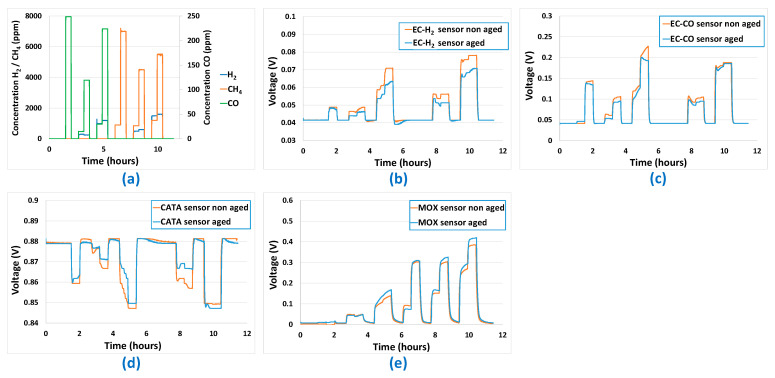
Comparison between the sensor responses before ageing and after one year of ageing: (**a**) Gas sequence used for the test, (**b**) Electrochemical EC-H_2_ sensor, (**c**) Electrochemical EC-CO sensor, (**d**) Catalytic CATA sensor, (**e**) Metal-Oxyde MOx sensor.

**Figure 18 sensors-24-03499-f018:**
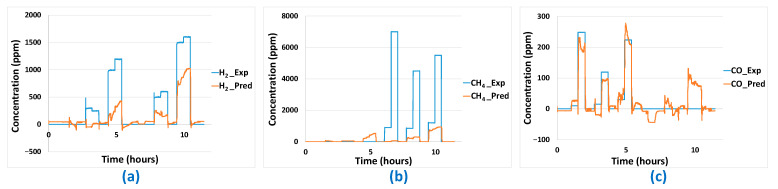
Gas concentration predictions (*_Pred* in the legend) on aged sensor platform for: (**a**) H_2_, (**b**) CH_4_, (**c**) CO using the previously developed ANN model—comparison to experimental used concentrations (*_Exp* in the legend).

**Figure 19 sensors-24-03499-f019:**
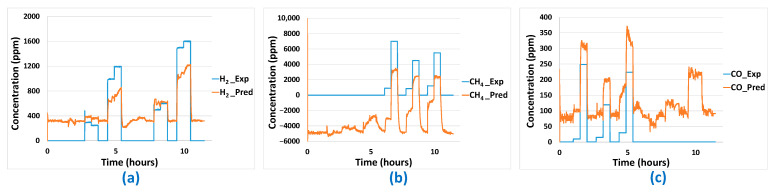
Gas concentration predictions (*_Pred* in the legend) on aged sensor platform for: (**a**) H_2_, (**b**) CH_4_, (**c**) CO using the previously developed MLR-PLS model—comparison to experimental concentrations (*_Exp* in the legend).

**Figure 20 sensors-24-03499-f020:**
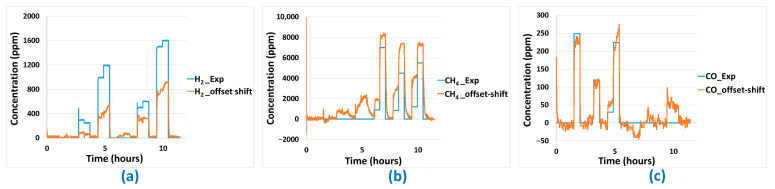
Offset-corrected gas concentration predictions (*_offset shift* in the legend) on aged sensor platform for: (**a**) H_2_, (**b**) CH_4_, (**c**) CO using the previously developed MLR-PLS model–comparison to experimental concentrations (*_Exp* in the legend).

**Table 1 sensors-24-03499-t001:** Technologies of sensors commercially available and corresponding to our specifications according to the target gas (X: Only a few commercial references found, XX: Many commercial references could be found but only a few corresponding to our requirements, XXX: many commercial references corresponding to our specifications found).

	CO	CO_2_	H_2_	CH_4_
** MOX (semiconductors) **	X		X	X
** Pellistors (Catalytic sensors) **			X	XX
** NDIR sensors **		XXX		X
** Photo-acoustic sensors **		X		
** Electrochemical **	XXX		X	

**Table 2 sensors-24-03499-t002:** List of selected sensors to be used in the sensor network cell.

	Brand Name	Model	Type	Detection Range	Detected Gas **
EC-CO	Membrapor	CO/MF-1000	Electrochemical	0–1000 ppm	CO
CATA	Figaro	TGS6812-D00	Catalytic	0–100% LEL *	H_2_, CH_4_, C_3_H_8_
MOX	Figaro	TGS2612-D00	Semiconductor	1–25% LEL *	H_2_, CH_4_, C_3_H_8_
CO_2_	Sensirion	SCD30	Infrared	0–40%	CO_2_ (+HR et T)
EC-H_2_	Membrapor	H2/M-4000	Electrochemical	0–4000 ppm	H_2_

* LEL stands for Lower Explosive Limit (which is the lowest concentration of a gas or vapor that will burn in air—about 4%, 5%, and 12.5% respectively for H_2_, CH_4_, and CO). ** according to the sensors’ datasheets.

**Table 3 sensors-24-03499-t003:** RMSE obtained with concentrations predicted from training data used to build the models.

Model	Training Prediction RMSE (ppm)		
	H_2_	CO	CH_4_
MLR—OLS	1801	895	11,060
MLR—PLS	66	35	656
Best ANN	103	34	671

**Table 4 sensors-24-03499-t004:** RMSE obtained with concentrations predicted from validation data.

Model	Training Prediction RMSE (ppm)		
	H_2_	CO	CH_4_
MLR—PLS	197	26	755
Best ANN	207	19	497

**Table 5 sensors-24-03499-t005:** RMSE obtained with concentrations predicted from validation data and post-treated.

Model	Training Prediction RMSE (ppm)		
	H_2_	CO	CH_4_
MLR—PLS	194	22	622
Best ANN	205	19	424

## Data Availability

The data that support the findings of this study are available from the corresponding author upon reasonable request.

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
