# Peer review of "Linear and Non-Linear Modelling Methods for a Gas Sensor Array Developed for Process Control Applications"

_sensors, 2024, doi:10.3390/s24113499_

Round 1
Reviewer 1 Report
Comments and Suggestions for Authors
The reviewed article addresses the important issue of process monitoring in "Power-to-X" area. The authors of this paper proposed the use of a sensor array to measure the concentrations of several gases associated with these technologies. The overall layout of the work, the course of the research, the results described and the conclusions are done correctly, but the weakness of the work is that it has little novelty.
In general, the subject of sensor arrays is nothing new, and the sensors used in the work do not feature any original solutions either. The same is true of the data processing algorithms used - the methods used can be considered very basic (which, however, does not disqualify them from performing well).
Here are some notes for consideration:
1) I suggest that the authors give more emphasis in the introduction to describing what makes their idea different from others found in the literature.
2) The description of the Power to X processes in Chapter 2.1. in which the sensor arrays could be applied is rather vague. It is not clear whether measurements of the concentrations of the described gases would take place in process reactors, in pipelines (under pressure) or in open spaces (at atmospheric pressure). Factors such as pressure and ambient humidity can have a significant impact on the performance of sensors (especially semiconductor sensors) - this topic is not addressed at all in the article.
3) What are the costs of purchasing the sensors and building the array? Perhaps the aspect of building a low-cost device would be another factor in justifying the purpose of the topic undertaken.
4) Authors should justify the choice of data processing methods used. The article does not refer to results from other publications in this area. It would have been useful to demonstrate that these fairly standard methods (MLR-OLS, MLR-PLS and ANN) give satisfactory results when measuring one- and two-component mixtures by means of gas sensor arrays.
5) The description of the gas mixture generation station and the sensor cell is very limited. It is not clear whether the gas mixtures flowed through this cell or whether they were introduced and retained there. I suggest posting a diagram of the test site or elaborating on the description.
6) The picture of the multi-sensors' cell in Figure 2 is no different from Figure 1 - Figure 1 seems unnecessary in this case
7) The titles above each diagram (Figures 5-20) seem redundant in view of the titles and descriptions under each figure.
8) Figures 9-11: parts a), b) and c) of the figure are not described in the title.
9) Figure 12: The lineage for CO is not very clear - I suggest this dataset be included on the second OY axis.
10) Figures 18-20: The abbreviations used in the legends are not very understandable.
Comments on the Quality of English Language
Some shortcomings were detected, e.g.:
* line 132: "egeing" - should be "ageing"
* inconsistency in naming: line 11 - Power to X, line 43 - "Power to X" (in quotation marks), line 81 - Power-to-X (with hyphens)
* inconsistency in separating units from numerical values, e.g. in Table 1 - ppm separated from the numbers; line 297, line 300, line 304, line 307 - ppm not separated from numbers
Reviewer 2 Report
Comments and Suggestions for Authors
The study addresses the need for process monitoring tools in new developments related to Power to X, focusing on H2, CO, CH4, and CO2 as key gases. To overcome the non-selectivity of sensors, the authors have constructed a multi-sensor matrix using commercial sensors with diverse transduction principles. This approach provides richer information for gas mixture composition determination (both nature and concentration). The authors have developed linear (Multi Linear Regression – Ordinary Least Square “MLR-OLS” and Multi Linear Regression – Partial Least Square “MLR-PLS”) and non-linear (Artificial Neural Network “ANN”) models to process the sensor array data. The MLR-OLS model was found ineffective during training and was disqualified for further use. Subsequently, the MLR-PLS and ANN models were evaluated using validation data. Both models demonstrated good concentration predictions for all analytes, with ANN performing better for methane due to the logarithmic response of MOX sensors to CH4 compared to the linear responses of other sensors to other analytes.
Comments that require addressing include the following:
The manuscript mentions the use of five commercial sensors with different transduction principles. It would be beneficial to elaborate on the specific differences between these sensors and the rationale behind selecting them for the study.
The basis for selecting H2, CO, CH4, and CO2 as the main gases of interest could be further explained.
The manuscript briefly discusses sensor drift in aged platforms. It would be helpful to expand on the factors that contribute to sensor drift, such as environmental conditions, sensor material degradation, or electronic components' aging.
Reviewer 3 Report
Comments and Suggestions for Authors
In this paper, the author proposes a linear and nonlinear modeling method for gas sensor arrays developed for process control applications, providing a favorable process monitoring tool for Power to X related process development. This article first constructs a multi-sensor platform to better detect the concentration of single and mixed gases, and then constructs linear and nonlinear models to better detect gases in mixtures. Although the work in this article has been fully introduced, there are still some issues that need to be improved and resolved in this article.
1. For Table 1, it is best for the content belonging to the same line table to be distributed on the same page. Regarding the arrangement of the position of this line table, the author is requested to reconsider.
2. For Figures 1 and 2, it is hoped that the author can reconsider the shooting equipment and environment, and beautify the sensor images to demonstrate the rigor and professionalism of the experiment.
3. The word "sensor network" in the first sentence of "2.1 Sensor's choice" may have ambiguity and can be replaced with other words
4. In the "3.1 Mono analyze tests" page, there are a large number of blank spaces below, which affects the overall structure of the paper. It is hoped that the author can fill this gap by adjusting the content of the paper.
5. In this article, the color of the chart text is gray, which may cause the chart text to be unclear in some cases. It is hoped that the author can replace it with clearer black for readers to read more easily.
6. For Figure 17, it is best for content belonging to the same chart to be distributed on the same page. Regarding the arrangement of the position of this chart, it is hoped that the author can reconsider.
7. Appendix A is not on the same page as the content, and we hope the author can make the necessary modifications.
8. The table in Appendix B has issues with stretching and layout, and there are also issues with the formatting and layout of the text in Appendix B. Please make the necessary modifications.

The overall language of the article is relatively smooth, but there are a few language issues that may lead to ambiguity. The author can further consider the vocabulary used in the article, in order to improve the quality of the language used in the article.
Round 2
Reviewer 1 Report
Comments and Suggestions for Authors
Dear authors,
thank you for your corrections. However, the reply letter and the revised manuscript did not refer to comment number 7). This comment is not particularly important, although the way these graphs are presented seems unprofessional (they look like they have been pasted directly from a spreadsheet).
In the case of the last comment (Comments on the Quality of English Language), my point was to standardise the presentation of the units, but precisely by separating them from the numerical values (such rules are generally accepted for the International System of Units).
